# BiEntropy, TriEntropy and Primality

**DOI:** 10.3390/e22030311

**Published:** 2020-03-10

**Authors:** Grenville J. Croll

**Affiliations:** Alternative Natural Philosophy Association, Bury St Edmunds IP30 9QX, UK; grenvillecroll@gmail.com

**Keywords:** prime number distribution, binary derivative, trinary derivative, Shannon entropy

## Abstract

The order and disorder of binary representations of the natural numbers < 2^8^ is measured using the BiEntropy function. Significant differences are detected between the primes and the non-primes. The BiEntropic prime density is shown to be quadratic with a very small Gaussian distributed error. The work is repeated in binary using a Monte Carlo simulation of a sample of natural numbers < 2^32^ and in trinary for all natural numbers < 3^9^ with similar but cubic results. We found a significant relationship between BiEntropy and TriEntropy such that we can discriminate between the primes and numbers divisible by six. We discuss the theoretical basis of these results and show how they generalise to give a tight bound on the variance of Pi(*x*)–Li(*x*) for all *x*. This bound is much tighter than the bound given by Von Koch in 1901 as an equivalence for proof of the Riemann Hypothesis. Since the primes are Gaussian due to a simple induction on the binary derivative, this implies that the twin primes conjecture is true. We also provide absolutely convergent asymptotes for the numbers of Fermat and Mersenne primes in the appendices.

## 1. Introduction

We developed the BiEntropy function [1] as a means of comparing the relative order and disorder of the digits of binary strings of arbitrary length. 

We originally tested the algorithm in the fields of Prime Number Theory, Human Vision, Cryptography, Random Number Generation, and Quantitative Finance. As a by-product of our work with prime numbers, we derived two very short corollaries which reaffirmed the irrationality of the prime constant [2].

We subsequently used BiEntropy to identify a significant difference between the alternating and non-alternating knots of 9 and 10 crossings [3] in the simple cubic lattice. Our work has been cited in the fields of cryptography, internet information processing, mobile computing, and random number generation [4,5,6,7,8]. Most recently, BiEntropy has been re-implemented, re-tested, and made publicly available on GitHub [9]. It has been prominently cited in a related US Government patent [10].

Despite this background of activity on the use and application of BiEntropy in diverse areas, including in particular prime number theory, we have failed until now to conduct the simplest of tests to ascertain if there was any relationship between BiEntropy and primality. The historical importance and deep roots of the problem of primality in mathematics finally caused us to commence our investigation. There is an extensive literature on prime number theory. Resources such as [11,12,13,14,15] and the references therein provide a useful background.

In this paper, we empirically investigate the relationship between BiEntropy and primality in the 8 and 32 bit binary strings. We then develop the TriEntropy function and investigate its relationship with primality within the 9 trit trinary strings. We briefly investigate the relationship between BiEntropy and TriEntropy. We conclude with a discussion of the theoretical basis behind this work and demonstrate how it generalises to all natural numbers.

All of the investigative, experimental, and computational work in this paper was performed within the Microsoft Excel spreadsheet environment [16]. This gave us great flexibility in the creative process, high development productivity, and notable computational and graphical functionality. These attributes have already been observed [17,18] and may facilitate the accessibility and furtherance of this work, especially within the educational domain [19,20]. 

The layout of this paper reflects the order in which the experimental and theoretical work took place, except that work on the Fermat and Mersenne primes was moved to Appendix C. We provide online a complete set of spreadsheets used to perform the computations and graphics within this paper. Details regarding access to these spreadsheets is given in the Appendix A.

## 2. BiEntropy

The BiEntropy algorithm uses a weighted average of the Shannon entropies [21] of a string and all but the last binary derivative [22] of the string. 

### 2.1. Shannon Entropy

Shannon’s Entropy of a binary string *s* = *s*_1_, …, *s_n_* where P(*s_I_* = 1) = *p* (and 0 log_2_ 0 is defined to be 0) is:
*H*(*p*) = −*p log*_2_*p* − (1 − *p*) *log*_2_ (1− *p*)(1)


For perfectly ordered strings which are all 1’s or all 0’s, i.e., *p* = 0 or *p* = 1, *H*(*p*) returns 0. Where *p* = 0.5, H(p) returns 1, reflecting maximum variety. However, for a string such as 01010101, where also *p* = 0.5, H(p) also returns 1, ignoring completely the periodic nature of the string. 

We can discover the periodicity of a binary string by using the binary derivatives of the string.

### 2.2. Binary Derivatives and Periodicity

The first binary derivative of *s*, d_1_(*s*), is the binary string of length *n* − 1 formed by XORing adjacent pairs of digits. We refer to the *k*th derivative of *s*, *d_k_*(*s*) as the binary derivative of *d*_*k*−1_(s). There are *n* − 1 binary derivatives of *s*. *p*(*k*) is the proportion of 1’s in *d_k_*. 

Almost fifty years ago, Nathanson [22], following the work of Goka [23], defined the notions of period and eventual period within arbitrary binary strings and outlined the related properties of binary strings and their derivatives both individually and collectively. Amongst a number of useful results, we find that a binary string is periodic with period 2*m* for some *m* ≥ 0 if and only if *d_k_*= 0 for some *k* ≥ 1.

### 2.3. BiEntropy Definition

BiEntropy, or BiEn for short, is a weighted average of the Shannon entropies of the string and the first *n* − 2 binary derivatives of the string. There are numerous ways of weighting the Shannon entropies. In this series of experiments, we weight the Shannon entropies using powers of two:
(2)$$\mathrm{BiEn}(s)=(1/(2^{n-1}-1))(\sum_{k=0}^{n-2}\;(-p(k)\cdot \log_{2}\;p(k)-(1-p(k))\cdot \log_{2}\;(1-p(k)))\cdot 2^{k})$$


The final derivative *d*_*n*−1_ is not used, as there is no variation in the contribution to the total entropy in either of its two binary states. The highest weight is assigned to the highest derivative *d*_*n*−2_. 

### 2.4. BiEntropy Properties

BiEntropy provides a number between 0 and 1 inclusive which indicates the relative order and disorder of the digits of a binary string of length *n* > 1. The shortest perfectly ordered strings are 00 and 11 which have a BiEntropy of 0. The only perfectly disordered strings are 01 and 10, which have a BiEntropy of 1. An ordered (i.e., periodic) string such as 01010101, for example, has a low BiEntropy of 0.01. A disordered string such as 10000110 has a high BiEntropy of, e.g., 0.95.

## 3. BiEntropy and Primality of the Natural Numbers < 256

We show in Figure 1 below the BiEntropy of the natural numbers < 256. The rows correspond to the most significant digits, and the columns to the least significant digits of their binary representations. The rows and columns are ordered by the 4 bit BiEntropy of the most and least significant digits, respectively. BiEntropy is colour coded with white < 0.15, yellow < 0.25, orange <0.5, and red < 1.0. Note the symmetry of the diagram about the diagonal. The primes are coloured purple. For example, 5 = 00000101 has an 8-bit BiEntropy of 0.23 and would be coloured yellow given the symmetry but is coded purple because it is (a Fermat) prime. The Fermat Prime 17 = 00010001 has a low BiEntropy of 0.05 due to the periodic nature of its digits. It would be coloured white but is coloured purple because of its primality. Furthermore, 127 = 01111111 has a BiEntropy of 0.92 and would be coloured red, however it is not only prime, but is a Mersenne Prime and is coloured purple. Note that 0 and 1 are simply “not prime”.

It is easy to see that most of the primes lie in the red quadrants, with only one prime (a Fermat prime) on the white diagonal. Note that the primality of the natural numbers < 256 has a variance with the natural symmetry of BiEntropy, as depicted in Figure 1.

The differences between the four prime proportions of Table 1 below are significant at *p* < 0.01. We have thus discovered a segmentation of the primes based upon BiEntropy, or more generally, the binary derivative. Looking for 8 bit primes in the red segment is approximately nine times more productive than looking in the white or yellow segments.

We show in Table 2 below the clear distinction between the BiEntropies of the primes, the non-primes, and the composite odd numbers at *p* < 0.0001 for the natural numbers < 256. The BiEntropy of the four Mersenne primes < 256 and the 33 twin primes < 256 is similar to the BiEntropy of all the Primes < 256. Thus, the number of the primes and the composite odds < 256 is 129, which includes the even prime.

If we sort the natural numbers < 256 by their BiEntropies and group them into eight segments as in Table 3, differences in prime density between the lowest and highest BiEntropy segment becomes markedly higher.

Prime density π(*x*), the number of primes less than or equal to *x,* is approximately *x*/ln (*x*) due to the Prime Number Theorems of Jacques Hadamard and Charles de la Vallée Poussin in 1896. BiEntropy appears to modify the prime density to ***O***(*x*^2^) for very small integers. Using BiEntropy or other prime density functions we can therefore usefully speak of *q*(*x*, *y*, *i*) which is the number of primes in the *i*th *y* sized ordered interval < *x*. Thus *q*(256, 32, 8) is 14, as above. Naturally, π(256) = *q*(256, 256, 1) = 54.

Finally, we depict the continuous relationship between BiEntropy and primality graphically in Figure 2, which reveals an almost deterministic relationship. We fit the related natural logarithm and quadratic curves and show the associated errors in Figure 3. We have adjusted the Natural Logarithm curve so that Log(256) matches π(256), which Li(*x*) does in the limit. Note that BiEntropy is a weighted average of the Shannon entropies of a binary string and the first *n* − 2 binary derivatives of the string. No (explicit) trial division has taken place in order to calculate BiEntropy. The number of primes < 256 = 54, and total BiEntropy for the primes < 256 = 42.64.

The means of Figure 3 are coincident due to the small multiplicative adjustment we made. The standard deviations of the errors are almost identical, at 0.93 for the natural logarithm and 0.98 for the quadratic. Thus, the actual error in the BiEntropic prime density for integers *x* < 256 is < √ *x* log(*x*) and is evidently Gaussian. As we will see, the error converges to 0 as *x*→ ∞.

## 4. BiEntropy& Primality of the Natural Numbers < 2^32^

### 4.1. Primes and Binary Derivatives

Whereas π(*x*)~*x*/ln(*x*), the number of binary derivatives used in the calculation of BiEntropy of a string of length *m* (where *m* = log_2_(*x*)) increases only as (*m*^2^− *m*)/2. We show in Table 4 the relationship between π(*x*), the number of primes, and the binary derivatives *d* for various *x*.

Thus, *d*/π(x) tends to 0 very rapidly, potentially rendering BiEntropy less sensitive to primality at longer string lengths.

### 4.2. Higher Powers of Shannon Entropy

There is some research originating in algorithmic information theory [24] that suggests that primality is related to disorder, which is of course what BiEntropy is designed to measure. This other work does not address the use of the binary derivative in this process. Note that there is only one prime on the diagonal of Figure 1, which is the region of maximum order.

We can change BiEntropy to sharpen its sensitivity to the detection of any departures from perfect disorder in the binary derivatives. This is trivially easy to do, especially within the spreadsheet environment, as we can simply and easily raise the Shannon entropy of each binary derivative to a power higher than 1.

We show in Figure 4 the effect of raising the powers of Shannon entropy from 1 to 10 based upon *p*, the variety. In the region of maximum variety in the middle of the chart where the variety, *p* = 0.5, the Shannon entropy is highest. When using higher powers of Shannon entropy, we can more powerfully discriminate departures from maximal disorder.

### 4.3. Investigating BiEntropy and Primality for x < 2^32^

We used a spreadsheet based Monte Carlo calculation to investigate a sample of natural numbers < 2^32^. Using a simple Excel data table, for each of 10,000 iterations, we generated a random 32 bit integer and then calculated its quadratic BiEntropy using the tenth power of the Shannon entropy of each derivative (P10 BiEntropy). We used a spreadsheet based exhaustive trial division calculation to determine the primality of each random 32 bit integer. We then sorted the sampled natural numbers and their BiEntropies into BiEntropic order and compared the prime density of this ordered interval with the natural prime density of the sample. We show the relationship between the sample’s natural and BiEntropic prime density in Figure 5 and the difference between the two densities in Figure 6.

### 4.4. Testing the BiEntropy and Primality Monte Carlo

We decided to carefully investigate the small difference between the natural and BiEntropic prime density generated by the Monte Carlo simulation, as depicted in Figure 5 above. The simulation consisted of 10,000 samples of an integer *x* in the range 0 < *x* < 2^32^ produced by the Excel RAND function, which we have previously scrutinised [1]. Since random numbers would be generated uniformly (i.e., linearly) in the given range, we were able to calculate, using the Prime Number Theorem, how many primes were likely to be produced during the generation of 10,000 random integers in the given range. We were then able to calculate a theoretical prime density for the Monte Carlo simulation to compare against the actual prime density of the Monte Carlo simulation.

We show the actual difference or Delta between the natural and BiEntropic prime densities with the theoretical expected difference or Delta in Figure 6. The theoretical Delta, shown in orange, accounts for only part of the difference. The difference between the BiEntropic prime density and the natural prime density is not accounted for by the difference between the linear production rate of prime numbers in the Monte Carlo simulation and the natural prime density. The difference is much larger. Squaring the expected difference and dividing by two (Delta^2^/2) matches the actual results of the Monte Carlo simulation much more closely. By examination, the error between theoretical and actual in Figure 6 is broadly normal (mean 1.22 and St. Dev. 6.17) and omitted for brevity. It appears to be the case that BiEntropic prime density is also quadratic for integers of ***O***(2^32^).

The number of primes actually produced in the Monte Carlo simulation we have reported is 391 compared with 473 expected prime numbers. A variation was to be expected.

## 5. TriEntropy

We have noticed in previous work that the BiEntropy function is not sensitive to periodicities of 3 (see the entries for 18, 27, 36, and 54 in Table A1 of Appendix B). For example, the 18 bit quadratic BiEntropy of 001001001001001001 is 0.9484 indicating disorder, however, the string is clearly periodic. We had thought that developing a trinary equivalent to BiEntropy might fix this problem but were not previously motivated to do so. Given the association between BiEntropy and primality outlined in the previous sections, and the fact that all primes ≥ 5 are of the form 6*k* ± 1, there became a clear motive to investigate.

### 5.1. Pairwise Addition and Differences Modulo 3

The acid test for TriEntropy was that it picked up periodicities of 3 within a trinary string. We devised a simple two way pairwise trinary addition table which we illustrate in Table 5.

We transformed our 8 bit binary BiEntropy calculator spreadsheet into a 9 trit TriEntropy calculator spreadsheet using the pairwise trinary addition table of Table 5 above. This took just a few minutes. Unfortunately, it did not work. We then discovered that in a 3 trit trinary string ABC we needed to compute the three-way pairwise trinary differences (PTD) between the three pairs AB, BC, AC, modulo 3. Thus,
*PTD* = *MOD*(*ABS*(*A* − *B*) + *ABS*(*B* − *C*) + *ABS*(*A* − *C*), 3)(3)
which we show in Table 6. It took a few more minutes in a spreadsheet to show that this did indeed work. The respective TriEntropies of the three trit strings did not look promising, however we persisted with our analysis. Note that the PTD function is invariant under pairwise permutation, none of A, B, or C having priority.

### 5.2. Computing TriEntropy

In order to compute the Shannon entropy of a trinary string, we need the *p_i_* of all the possible symbols. For the derivatives, as in Table 6 above, the *p_i_* for 0, 1, 2, are 0.111 (3/27), 0.444 (12/27), and 0.444 (12/27), respectively. Importantly, since TriEntropy would necessarily calculate the Shannon entropy of the original string, note that the *p_i_* of the 0,1,2 of the input string are 0.333, 0.333, 0.333, as they are equiprobable. Furthermore, note that only (*n* − 1)/2 − 1 derivatives are possible (where *n* is odd) as three input trits are required to compute each output trit of the derivatives. Finally, note that in BiEntropy, once a periodicity is detected, the further derivatives automatically fall to 0. This is not the case for TriEntropy, hence, derivatives that fall to 0 must have their further non-use programmed in specifically. Note *n* is odd.
(4)$$\mathrm{TriEn}(s)=(1/(\sum_{k=0}^{(n-1)/2}3^{k}))(\sum_{k=0}^{(n-1)/2}(-p(k)\cdot \log_{2}\;p(k)-(1-p(k))\cdot \log_{2}\;(1-p(k)))\cdot 3^{k})$$


We show in Table 7 the layout of a simple Excel spreadsheet to compute the polynomial (i.e., cubic) TriEn of a 9 trit string. We used Table 7 to compute each trit of the derivatives.

We exhaustively calculated TriEn for all *x* < 3^9^ and show the resulting natural and TriEntropic prime densities in Figure 7. In the equivalent BiEntropy diagram, for all *x* < 2^16^, BiEntropy is almost identical and was earlier omitted for brevity. We show the difference or Delta between TriEntropic prime density and natural prime density in Figure 8. 

Thus, the difference between the natural and TriEntropic prime densities for *x* < 3^9^ is approximately cubic. The error of the difference is approximately Gaussian, which we depict in Figure 9. The mean error is 0.00 with a standard deviation of 7.34.

## 6. Interaction between BiEntropy and TriEntropy

We investigated the interaction between BiEntropy and TriEntropy in the natural numbers < 256. We did this by allocating two segment numbers between 0 and 15 inclusive to each natural number depending upon the BiEntropy and TriEntropy. The 16 natural numbers with the lowest BiEntropy were allocated to BiEntropy segment 0 and the 16 natural numbers with the highest BiEntropy were allocated to BiEntropy Segment 15, etc., and similarly for TriEntropy. We show in Figure 10 below, a diagram of the frequency of occurrence of the primes in blue and numbers divisible by six in red arranged by BiEntropic segment number on the *x* axis and by TriEntropic segment number on the *y* axis. The primes are coded as positive numbers and the numbers divisible by six are coded as negative numbers. There was one collision in segment 8–9 corresponding to the numbers 42 and 103 which is coded yellow.

Although the data volume is small, we expect from our earlier experiments that increasing BiEntropy and increasing TriEntropy will disclose more primes and fewer composites. This is what appears to be the case. Ignoring the bottom left to top right diagonal, primes are relatively absent from the top left triangle (11/120 versus 40/120, *p*< 0.0001) and numbers divisible by six are relatively absent from the bottom right triangle (11/120 versus 30/120, *p*< 0.002), which corresponds to prior expectation. There was only one segment collision, whereas eight might have been expected (54 * 42/256) if the distribution of primes and numbers divisible by six was uniform across all the BiEntropic and TriEntropic segments. Note that the 202 non-primes are uniformly distributed across Figure 10, which information is not shown for brevity, but is available in the Appendix A.

## 7. Theoretical Basis

### 7.1. Introduction

We now show why the notion of periodicity is so critically important in the determination of primality.

### 7.2. Periodic and Non-Periodic Numbers

Consider the concatenated binary string *ab* where the length of *a* and *b* is *n* and *n* ≥ 1, so the length of *ab* is 2*n*. If *a* = *b* for some *n*, then *ab* is periodic. The periodic numbers appear along the diagonal emanating from the origin of Figure 1 (where *n* = 4) and are mostly coloured white.

### 7.3. Periodic Binary Primes

Where *a* = *b* = 1, some of these are the Fermat numbers, of which only five are known to be prime [25]. A Fermat number, 17, appears on the diagonal of Figure 1 and is coloured purple because it is prime. We discuss the Fermat numbers in more detail in Appendix C.

### 7.4. Periodic Binary Composites

The rest of the numbers, *k*, on the diagonal emanating from the origin of Figure 1 and its equivalents for all *n* are of the form:
*k* = (2^*n*^ ∙ *a*) + *b*(5)
since *a* = *b*then
*k* = (2^*n*^ + 1) ∙ *a*(6)
since *a* > 1therefore, *k* is composite. 


The first periodic binary composite > 0 is 1010, which is 10 (ten). Thus, the Mersenne numbers (numbers of the form 2*^n^*− 1) of even length (where *a = b*) cannot be prime. The odd length Mersenne numbers, e.g., 0111, seven, may be prime but are not periodic because *a ≠ b*. We list the periodic binary composites < 256 in Appendix B and discuss the Mersenne numbers in more detail in Appendix C.

### 7.5. N-Periodic Binary Composites

Numbers of the form 00111100 and 10010110, etc., where *a* is the 2’s complement of *b*, i.e.,
*b* = 2^*n*^− *a* − 1(7)
are also composite. These numbers appear in white in the short cross diagonals of Figure 1. These numbers are also of the form:
*k* = (2^*n*^ ∙ *a*) + *b*(8)

substituting
*k* = (2^*n*^ ∙ *a*) + 2^*n*^ − *a* − 1(9)

therefore,
*k* = *a* ∙ (2^*n*^− 1) + 2^*n*^− 1(10)
*k* = (*a* + 1) ∙ (2^*n*^− 1) (11)
if *a* ≥ 1 then *k* is compositeelse *a* = 0 and *b* is a periodic binary composite (e.g., 1111…) of length *n*/2


### 7.6. Periodic M-Ary Primes

All primes *k* > 2 are of the form *ab* where *a* and *b* are of length *n* and *a = b = 1* in one base *m* = *k* − 1.
since a = b = 1then
*k* = ((*k* − 1)^1^ ∙ 1) + 1(12)



Except for the Fermat primes, which are also periodic in base 2.

That is,
*k* = (2^(*n* − 1)^ · *a*) + *b* where *a* = *b* = 1.(13)


### 7.7. Periodic M-Ary Composites

In general, the numbers *k*, on the diagonals of diagrams equivalent to Figure 1 in any base *m* are of the form:
*k* = (*m^n^* ∙ *a*) + *b*(14)
since *a* = *b*then
*k* = (*m^n^* + 1) ∙ *a*(15)
since *a* > 1, *k* is composite. 


### 7.8. Non Periodic Numbers

Numbers where *a* ≠ *b* (*n* ≥ 1) in any base are either prime or non-prime.

## 8. Discussion

Thus, the principal reason why BiEntropy and TriEntropy have any relationship with primality is the simple fact that, except for the Fermat numbers (and e.g., 23 = 11_22_), periodic and *n*-periodic numbers cannot be prime in any base. Hence, the main diagonal of Figure 1 (for all *x*, and in all bases) is almost devoid of primes and there are no primes on the cross diagonals. Ignoring the Fermat primes, 32/256 = 12.5% of natural numbers < 256 cannot be prime due to the periodicity or *n*-periodicity within seven of their last eight binary derivatives.

If a binary string is periodic, one, and then all the further derivatives, fall to 0 [22]. BiEntropy picks this up, as the Shannon entropy is 0. Symmetrically, if a derivative is all 1’s, it will also have a 0 Shannon entropy and will (unless it is the last used derivative) become all 0’s in the next derivative. The earlier that periodicity is observed (i.e., for shorter periods), the lower the weighted total becomes, as all the higher weights are 0. Non-periodic strings are otherwise ranked accordingly, with those strings with the most derivatives at or close to *p* = 0.5, gaining the highest BiEntropy. BiEntropy is the Hamming distance for primality. Except in certain circumstances (e.g., *s* = 00000001), the bits of a binary derivative are undecidable. Determination of the bits of a binary derivative is a simple variation of the halting problem—if the last binary derivative is 1, the routine halts else it does not halt.

Whilst a string is periodic if and only if one of its derivatives is all 0’s [22], the reverse does not apply, hence primality is stochastic. Davies et al. [26] proved that if the bits of a string occur with a probability of 0.5, the bits of the derivatives also occur with a probability of 0.5, and the binary derivatives are independent. Hence, the error between a quadratic and the BiEntropic prime density, which is a quadratic function of the probability of occurrence of each bit of the derivative, is Gaussian due to the central limit theorem. Note that the number of binary derivatives for any *x* is finite.

By a simple induction, every binary number is the binary derivative of a number one bit longer, and its bits occur with a probability of 0.5. Its bits are proven [26] independent of its earlier derivatives. Hence, the primes are Gaussian, as the probability of occurrence of each of their bits is undifferentiated from every other binary number and every other binary derivative.

BiEntropic prime density is quadratic because BiEntropy is quadratic. For example, in the 8 bit version of BiEntropy, the probability of an arbitrary string not being prime because the string is all 1’s, or the probability that a binary derivative is 0, is:
P(*s* is not prime) = 1/256+ 1/256 + 2/256 + 4/256 + 8/256 + 16/256 + 32/256 + 64/256 = 128/256 = 0.5(16)
i.e., there is only one 8 bit string (*s* = *d*_0_) that is all 0’s, whereas the last used derivative *d*_6_ of length 2 is all 0’s on 64 occasions and *d*_5_ of length 3 is all 0’s on 32 occasions. BiEntropy measures exactly the probability that a binary string cannot be prime or may be prime with a precision given by the number of bits in *d*_0_. TriEntropy is cubic for similar reasons.

The relationship between BiEntropy and primality generalises for all *x* for the simple reason that all *x* ≥ 256 (for example) eventually end up as an 8 bit (for example) string by virtue of successive binary differentiation. Determination of many of the mathematical and statistical properties of all *x* can be obtained inductively by observation of properties in the last *m* binary derivatives, which is easy to do when *m* is small.

Thus, there exists a set of constants *a_k_*, *b_k_*, and *c_k_*, such that
*a_k_*.*x_k_*^2^ + *b_k_*.*x_k_* + *c_k_* = *Li*(*x_k_*) where *x_k_* = *m*^2^, *m* is integer and *c_k_* = 0(17)


And another (similar) set of constants *u_k_*, *v_k_*, and *w_k_*, such that
*u_k_*.*x_k_*^2^ + *v_k_*.*x_k_* + *w_k_* = *π*(*x_k_*) where *x_k_* = *m*^2^, *m* is integer and *w_k_* = 0(18)


For each *a_k_*, *b_k_*, *c_k_* and *u_k_*, *v_k_*, and *w_k_*, there exists a set of (*m*^2^ − *m*)/2 binary derivatives from which the distribution of primes is derived with known probabilities and calculable or estimable variance. The variance in natural prime density is constrained by the variance in the BiEntropic prime densities for all *x_k_* < *x* because the same data—the natural numbers—is Gaussian distributed about two differing central measures—a quadratic and a logarithmic integrand.

Since
Lim *π*(*x*)/*Li* (*x*) ~ 1 *x*→∞(19)


Therefore, in the limit, the BiEntropic/Quadratic and Logarithmic Integrand/Natural error distributions are coincident with near identical error distributions, which we illustrated empirically in Figure 3. Furthermore, as *x*→∞, and since the number of bits in the binary derivatives = (*m*^2^ − *m*)/2, where *m* = log_2_(*x*), the variance in the error between the BiEntropic and Quadratic prime densities is ***O***(log(*x*)/*x*) due to the central limit theorem. Hence, the error between the Logarithmic Integrand and the natural prime densities rapidly tends to 0.
i.e., Lim Var(*π*(*x*) − *Li* (*x*)) → 0*x*→∞(20)
which is clearly distinctive from the von Koch [27] bound for proof of the Riemann Hypothesis.
*π*(*x*) − *Li* (*x*) = ***O*** (√ *x* log(*x*))(21)


A similar set of cubic constants apply for TriEntropy and the arithmetic addition of BiEntropy and TriEntropy, which we shall denote TriBiEntropy. We illustrate the cubics of TriBiEntropy intersecting *π*(*x*) for various *x* in Figure 11.

## 9. Conclusions

We have shown a clear empirical link between BiEntropy and primality for the natural numbers < 2^8^. We have repeated this analysis statistically for the natural numbers < 2^32^ and found similar results, including the prime density remaining ***O***(*x*^2^). We developed a related TriEntropy function and showed that TriEntropy changes prime density to ***O***(*x*^3^) for the natural numbers < 3^9^. In addition, TriEntropy has addressed a natural weakness in the detection of periods of length 3, or multiples thereof, in the BiEntropy function.

Since BiEntropy and TriEntropy are simply measures of the order and disorder (i.e., the periodicity) of a string, the implication is that prime numbers expressed in binary or trinary have more disordered representations. The reverse implication is that composites have more ordered representations. This result has been suggested in earlier work in algorithmic information theory.

We have shown how to increase the sensitivity of BiEntropy by increasing the exponent of the Shannon entropy within the BiEntropy calculation. We have demonstrated a significant link between BiEntropy and TriEntropy in the natural numbers < 2^8^ and the practicality of combining BiEntropy and TriEntropy via arithmetic addition. We have given a brief outline of the theoretical underpinnings behind this initial experimental work and shown how it generalises for all the natural numbers.

We have shown how the variance of the error between *π*(*x*) and *Li* (*x*) tends to 0 due to the Gaussian constraints on the variance of *π*(*x*) imposed by the binary derivative. These are much tighter constraints than the bound proven by Von Koch in 1901 as being equivalent to proof of the Riemann Hypothesis.

We have provided, in Appendix C, easily derived absolutely convergent asymptotes for the numbers of Fermat and Mersenne primes.

Finally, since the distribution of primes is Gaussian due to the binary derivative, this implies that the twin primes conjecture is true.

## 10. Further Work

Note from Figure 1 and earlier work, that BiEntropy, TriEntropy, etc., though reals are quantized and not continuous and have a finite number of states. This may be of relevance in attempts to relate BiEntropy and TriEntropy to the physical domain [28].

There are myriad opportunities to relate primality to other domains [29], particularly bearing in mind the now established connections between binary, the binary derivative, primality, and their *m*-ary generalisations.

## 11. Patents

The BiEntropy function is prominently cited in Gurrieri, T.M., Hamlet, J.R., Bauer, T.M., Helinski, R., & Pierson, L.G. (2018), US Patent for Integrated circuit physically unclonable function (US Patent # 10,103,733) [10]

## Figures and Tables

**Figure 1 entropy-22-00311-f001:**
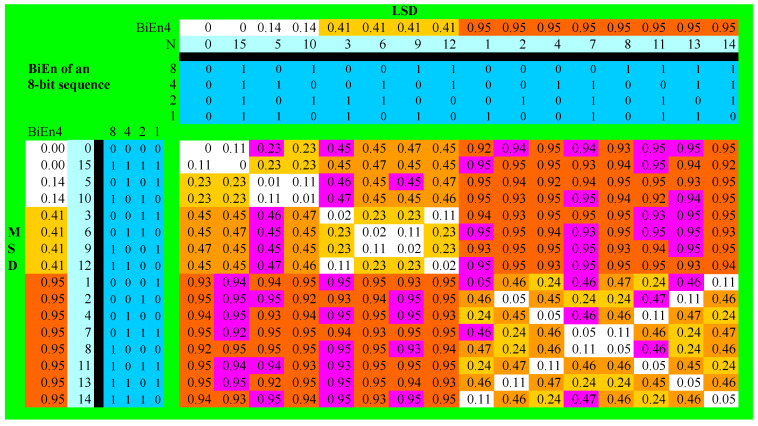
BiEntropy and primality of the Natural Numbers < 256.

**Figure 2 entropy-22-00311-f002:**
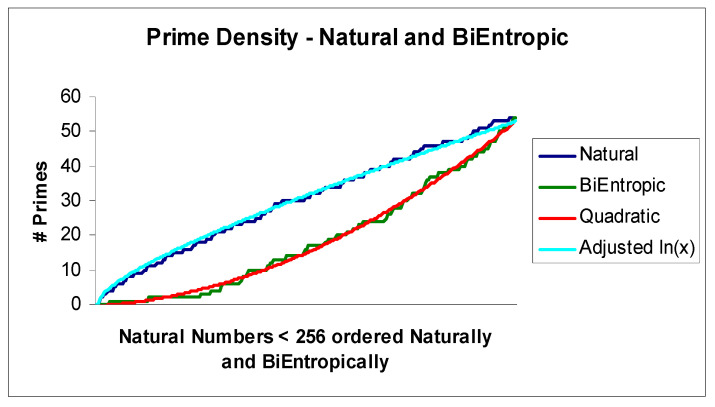
BiEntropy modified prime density.

**Figure 3 entropy-22-00311-f003:**
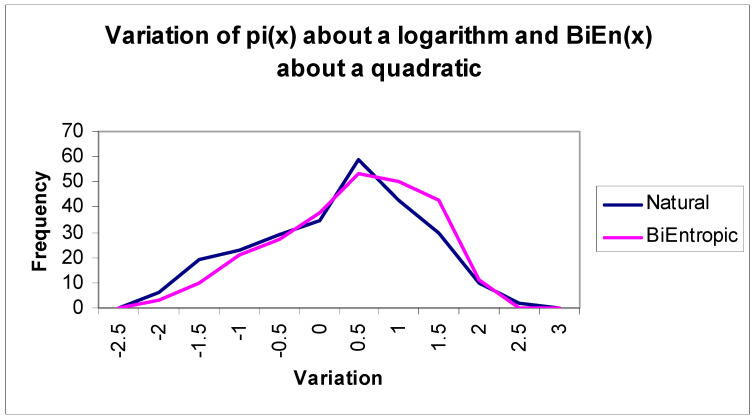
Variation in prime density.

**Figure 4 entropy-22-00311-f004:**
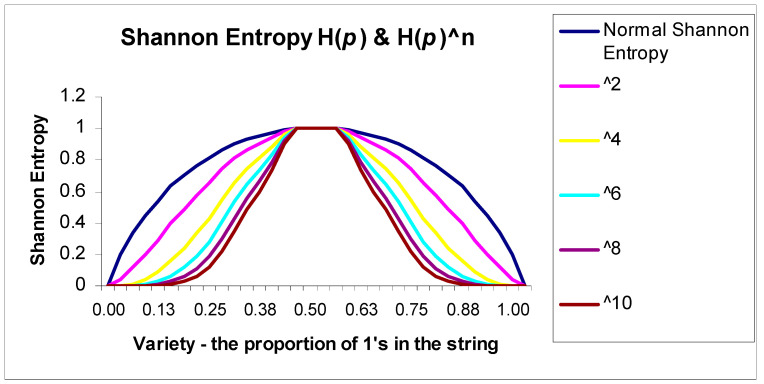
Raising Shannon entropy to a higher power.

**Figure 5 entropy-22-00311-f005:**
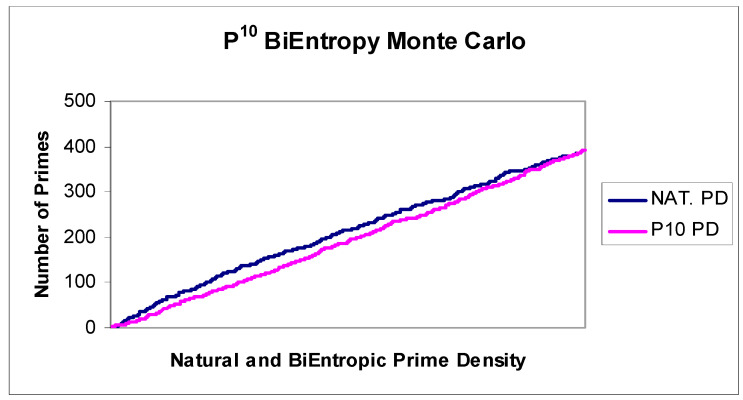
P10 BiEntropy and prime density.

**Figure 6 entropy-22-00311-f006:**
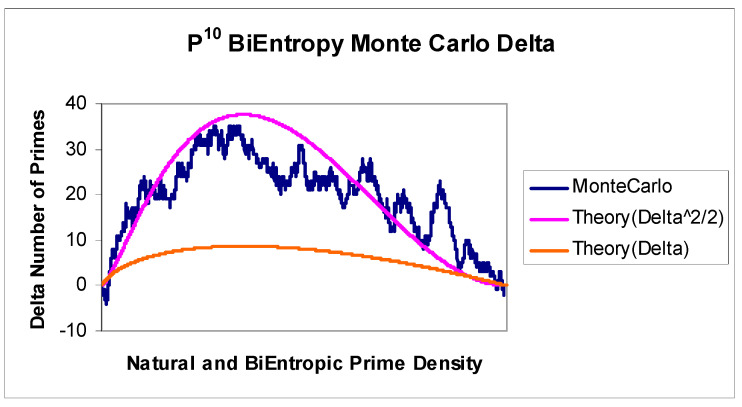
P10 BiEntropy and prime density Delta.

**Figure 7 entropy-22-00311-f007:**
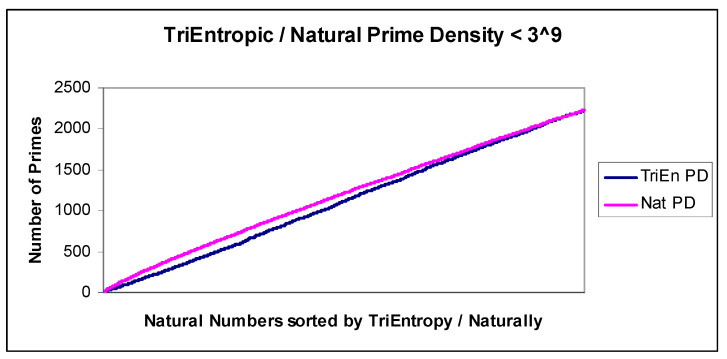
TriEntropy and prime density.

**Figure 8 entropy-22-00311-f008:**
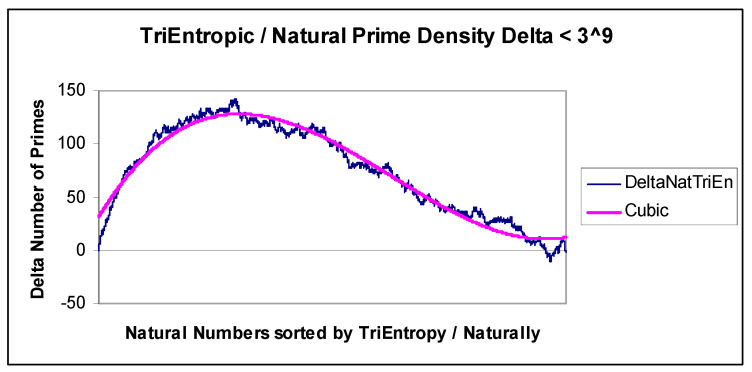
TriEntropy and prime density Delta.

**Figure 9 entropy-22-00311-f009:**
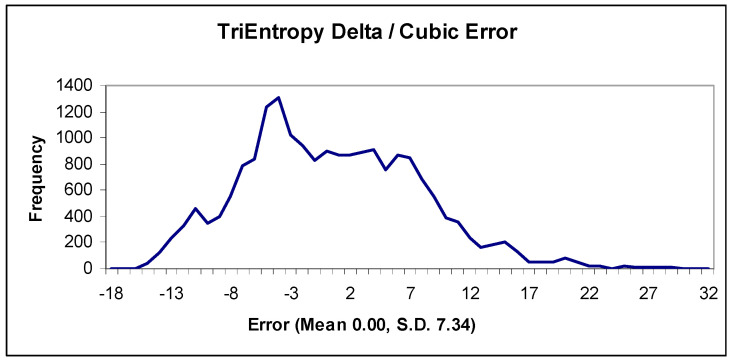
TriEntropy Delta/cubic error.

**Figure 10 entropy-22-00311-f010:**
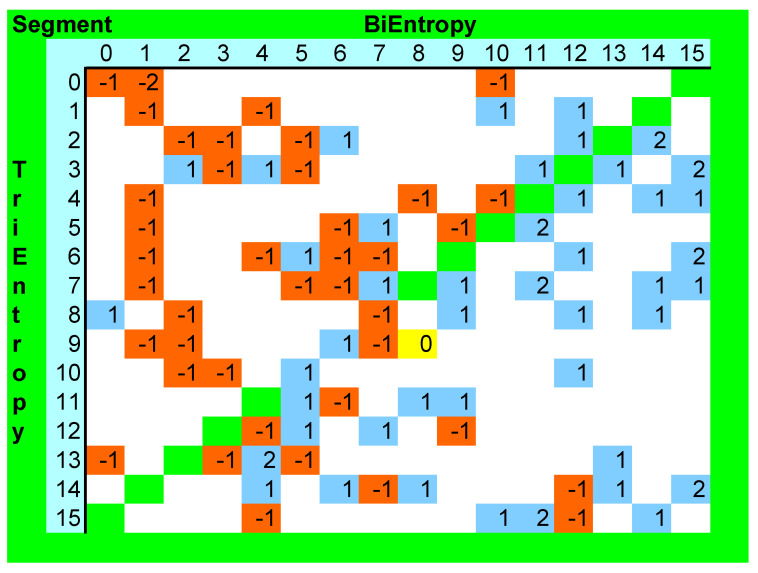
BiEntropy and TriEntropy interaction < 256.

**Figure 11 entropy-22-00311-f011:**
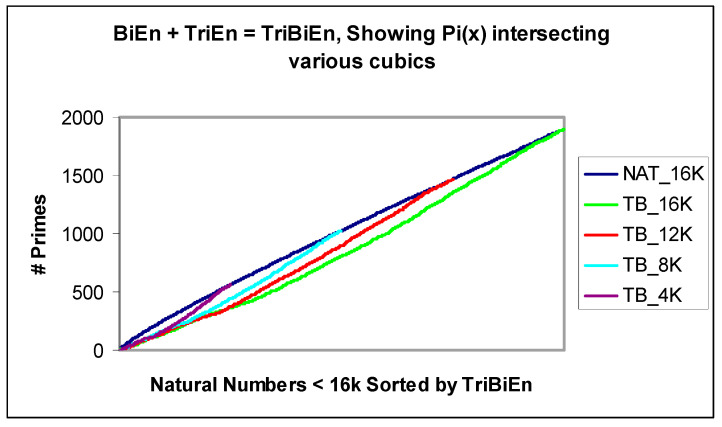
BiEntropy + TriEntropy and π(*x*) for various *x* < 16k.

**Table 1 entropy-22-00311-t001:** Prime Proportions.

Colour Code	BiEntropy	Count	Prime	Prime Proportion
White	<0.15	32	1	0.0312
Yellow	<0.25	32	1	0.0312
Orange	<0.50	64	15	0.2343
Red	<1.00	128	37	0.2890

**Table 2 entropy-22-00311-t002:** Mean BiEntropy.

	Prime	Not Prime	Odd	Mersenne	Twin
Mean	0.7897	0.5863	0.5099	0.8134	0.7783
S.Dev	0.2505	0.3444	0.3497	0.2443	0.2674
N	54	202	75	4	33

**Table 3 entropy-22-00311-t003:** BiEntropy ordered prime segments.

Segment	BiEntropy ≤	Primes
1	0.1141	1
2	0.2395	1
3	0.4558	8
4	0.4734	6
5	0.9350	6
6	0.9487	9
7	0.9506	9
8	0.9532	14

**Table 4 entropy-22-00311-t004:** π(x) and the number of binary derivatives for various x.

*x*	Bits(*m*)	π(x)	Derivatives(*d*)	*d*/*π*(*x*)%
256	8	54	28	51.85%
65536	16	6542	120	1.83%
4,294,967,296	32	203,280,221	496	0.00%

**Table 5 entropy-22-00311-t005:** Pairwise trinary addition table.

	**0**	**1**	**2**
**0**	0	1	2
**1**	1	0	1
**2**	2	1	0

**Table 6 entropy-22-00311-t006:** Pairwise trinary difference (PTD) Table.

A	B	C	PTD	TriEntropy
0	0	0	0	0.168
0	0	1	2	0.395
0	0	2	1	0.395
0	1	0	2	0.395
0	1	1	2	0.395
0	1	2	1	0.395
0	2	0	1	0.395
0	2	1	1	0.395
0	2	2	1	0.395
1	0	0	2	0.395
1	0	1	2	0.395
1	0	2	1	0.395
1	1	0	2	0.395
1	1	1	0	0.168
1	1	2	2	0.395
1	2	0	1	0.395
1	2	1	2	0.395
1	2	2	2	0.395
2	0	0	1	0.395
2	0	1	1	0.395
2	0	2	1	0.395
2	1	0	1	0.395
2	1	1	2	0.395
2	1	2	2	0.395
2	2	0	1	0.395
2	2	1	2	0.395
2	2	2	0	0.168

**Table 7 entropy-22-00311-t007:** Computing 9-Trit TriEntropy.

Trinary Expansion of N	Len(N)	N0	N1	N2	p	1−p	−p.log(p)	−(1−p).log(1−p)	TriEn	k	3^k	TriEn.3^k
111201101	9	2	6	1	0.33	0.67	0.53	0.39	0.92	0	1	0.92
0211222	7	1	2	4	0.40	0.60	0.53	0.44	0.97	1	3	2.91
12220	5	1	1	3	0.38	0.62	0.53	0.43	0.96	2	9	8.61
201	3	1	1	1	0.33	0.67	0.53	0.39	0.92	3	27	24.79
									3.76	6	40	37.23
									TriEn(s)			0.93

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
