# Peer review of "BiEntropy, TriEntropy and Primality"

_entropy, 2020, doi:10.3390/e22030311_

Round 1
Reviewer 1 Report
The paper has a great significance in both finding the connection between BiEntropy and TriEntropy and providing a solution in the form of spreadsheet application available to a wide range of end-users. The author show merit in the combination of theoretical mathematics and a simple end-user spreadsheet interface/tool. He proves a theory connected to both mathematics and computer sciences using the simplified functional language of spreadsheet applications, and applies its simple graphical tools to demonstrate and explain the results. With this methodology, beyond the importance of the proved hypothesis, the author provides an excellent example how non-professional programmers can apply simple programming tools in problem-solving of various fields. This is an excellent paper.
Author Response
Dear Reviewer 1,
Thank you for your review.
There are no specific points from yourself to respond to.
Regards
Grenville Croll
Reviewer 2 Report
In the paper under consideration, the author studies the interesting connection between BiEntropy and Primality in 8 and 32 bit binary strings. Furthermore, he develops the TriEntropy function and studies its connection to primality within the 9 trit trinary strings. Within his investigation, the author finds an interesting relationship between BiEntropy and TriEntropy which is such that one can discriminate between primes and numbers which are divisible by 6. The author also discusses how the findings of his study could also be generalized for all positive integers as well.
Overall the paper is well written and the content is interesting and has a nice interdisciplinary flavor. Due to the historical importance and deep roots of the problem of Primality in Mathematics, I would strongly recommend to the author to present in his Introduction some further relevant information to prime number theory as well assist the reader to find further sources by enhancing the literature of the paper by presenting more books and papers relevant to this domain. Some such sources, presenting information related to Analytic Number Theory (primality, Riemann Hypothesis, Fermat and Mersenne primes, etc) are the following (as well as the literature therein):
1) R. Crandall and C. Pomerance, Prime Numbers: A Computational Perspective, Springer, 2005
2) R. Guy, Unsolved Problems in Number Theory, Springer, 2004.
3) M. Th. Rassias, Problem-Solving and Selected Topics in Number Theory, Springer, 2011.
4) S. Miller and R. Takloo-Bighash, An Invitation to Modern Number Theory, Princeton Univ. Press, 2006
5) J. F. Nash, Jr. and M. Th. Rassias, Open Problems in Mathematics, Springer, 2016
Additionally, it would be nice to also mention some other works which are of strong interdisciplinary nature and which associate the ancient problem of the study of primes with other very modern fields, such as:
6) E. Guariglia, Primality, Fractality, and Image Analysis, Entropy, 21(3)(2019), 304. https://doi.org/10.3390/e21030304
The above would certainly be useful to the readers and will assist as well as encourage them to possibly contribute in this domain as well.
On the basis of the above minor revision, I recommend the paper to be published in Entropy.
Author Response
Dear Reviewer 2,
Thank you for your detailed review. I accept your recommendation to improve the introduction and references regarding the historical background and modern connections. Thank you for the suggested list of books and papers.
Regards
Grenville Croll
Reviewer 3 Report
Accept after minor revision (corrections to minor methodological errors and text editing)
Author Response
Thank you for your review.
Please provide some detail regarding your requests for minor revision.
Regards
Grenville Croll